# First Report of *Candida auris* Candidemia in Portugal: Genomic Characterisation and Antifungal Resistance-Associated Genes Analysis

**DOI:** 10.3390/jof11100716

**Published:** 2025-10-03

**Authors:** Isabel M. Miranda, Micael F. M. Gonçalves, Dolores Pinheiro, Sandra Hilário, José Artur Paiva, João Tiago Guimarães, Sofia Costa de Oliveira

**Affiliations:** 1Rise-Health Unit, Department of Surgery and Physiology, Faculty of Medicine of the University of Porto, Alameda Prof Hernani Monteiro, 4200-319 Porto, Portugal; imiranda@med.up.pt; 2Centre for Environmental and Marine Studies (CESAM), Department of Biology, University of Aveiro, 3810-193 Aveiro, Portugal; mfmg@ua.pt; 3Department of Clinical Pathology, São João University Hospital, Alameda Prof Hernani Monteiro, 4200-319 Porto, Portugal; m.dolores.pinheiro@gmail.com (D.P.); jtguimar@med.up.pt (J.T.G.); 4GreenUPorto—Sustainable Agrifood Production Research Centre/Inov4Agro, Department of Geosciences, Environment and Spatial Plannings, Faculty of Sciences, University of Porto, 4485-646 Vila do Conde, Portugal; hilario.sandra@fc.up.pt; 5Intensive Care Medicine Service, Local Health Unit Sao Joao, Faculty of Medicine of the University of Porto, Alameda Prof Hernani Monteiro, 4200-319 Porto, Portugal; jarturpaiva@gmail.com; 6Epidemiology Research Unit (EPIUnit), Department of Biomedicine, Faculty of Medicine, Institute of Public Health, University of Porto, Alameda Prof Hernani Monteiro, 4200-319 Porto, Portugal; 7Rise-Health Unit, Department of Pathology, Faculty of Medicine of the University of Porto, Alameda Prof Hernani Monteiro, 4200-319 Porto, Portugal

**Keywords:** *Candida auris*, candidemia, invasive, colonisation, antifungal resistance, *CRZ1* mutation

## Abstract

*Candida auris* has emerged as a global public health threat due to its high mortality rates, multidrug resistance, and rapid transmission in healthcare settings. This study reports the first documented cases of *C. auris* candidemia in Portugal, comprising eight isolates from candidemia and colonised patients admitted to a major hospital in northern Portugal in 2023. Whole-genome sequencing (WGS) was performed to determine the phylogenetic relationships of the isolates, which were classified as belonging to Clade I. Genome sequencing also enabled the detection of missense mutations in antifungal resistance genes, which were correlated with antifungal susceptibility profiles determined according to EUCAST (European Committee on Antimicrobial Susceptibility Test) protocols and guidelines. All isolates exhibited resistance to fluconazole and amphotericin B according to the recently established EUCAST epidemiological cut-offs (ECOFFs). Most of the isolates showed a resistant phenotype to anidulafungin and micafungin. All isolates were resistant to caspofungin. Missense mutations identified included Y132F in *ERG11*, E709D in *CDR1*, A583S in *TAC1b*, K52N and E1464K in *SNQ2*, K74E in *CIS2*, M192I in *ERG4*, a novel mutation S237T in *CRZ1*, and variants in *GCN5*, a gene involved in chromatin remodelling and stress-response regulation. Identifying known and novel mutations highlights the evolution of antifungal resistance mechanisms in *C. auris*. These findings underscore the need for further research to understand *C. auris* resistance pathways and to guide effective clinical management strategies.

## 1. Introduction

*Candida auris* classification as a global public health threat is underscored by its epidemiological data, which reveals a widespread distribution across continents, affecting patients in over 60 countries [1,2]. This pathogen is associated with high mortality rates, and around 60% of patients infected may die from it, particularly in vulnerable populations in intensive care units (ICU) [2,3]. The rapid emergence and dissemination of *C. auris* highlight a urgent need for enhanced global surveillance and improved diagnostic capabilities. In 2022, the World Health Organisation classified this fungus as a priority pathogen due to its rapid transmission in healthcare settings and high mortality rate [4,5].

*C. auris* belongs to the *Candida auris-Candida haemuli* clade (CAH) from the family *Metschnikowiaceae* [6]. However, recent phylogenomic and comparative genomics analyses have proposed a reassignment to the genus *Candidozyma* [7]. The phylogenetic placement of this group has been challenging because members of the CAH clade share high sequence similarity in ribosomal markers, display convergent phenotypic traits, and appear to have undergone a relatively recent diversification. As a result, traditional phylogenetic markers often lack resolution, and only genome-scale analyses have provided sufficient phylogenetic signal to support their reassignment to *Candidozyma* [7].

To date, six geographically distinct *C. auris* clades have been described, including Clade I (South Asian), Clade II (East Asian), Clade III (South African), Clade IV (South American), Clade V (Iranian) and Clade VI (Singapore and Bangladesh) [1,8,9,10]. Clade-specific characteristics provide essential insights into *C. auris* epidemiology and clinical behaviour. Clades I, III, and IV are closely associated with outbreaks of invasive and multidrug-resistant infections, highlighting their significant role in healthcare-associated transmission [11]. In contrast, Clade II strains often display susceptibility to all antifungal agents and are predominantly linked to ear infections rather than invasive diseases. Clade II exhibits unique genomic features, such as distinct karyotypes characterised by extensive subtelomeric deletions and chromosomal rearrangements. This may contribute to its reduced propensity for causing systemic infections [12]. The clinical relevance of Clades V and VI is not yet fully understood due to limited studies so far.

Antifungal resistance mechanisms in *C. auris* isolates are multifaceted and involve significant fitness trade-offs, often compensated by adaptive mechanisms through regulatory network adaptations and increased stress tolerance that enable persistence and pathogenicity [13]. These compensatory adaptations highlight *C. auris* remarkable ability to balance the costs of antifungal resistance with fitness requirements, contributing to its persistence in healthcare and environmental settings [14].

In Portugal, a single case of *C. auris* belonging to Clade III was identified in 2023 in a colonised patient transferred from Angola, admitted to an ICU [15]. Our study documents the first reported cases of *C. auris* candidemia identified in 2023, highlighting the need for urgent surveillance and infection control measures. The primary objective was to characterise eight epidemiologically linked *C. auris* isolates obtained from candidemia and colonised patients admitted to the largest hospital in northern Portugal, focusing on their genetic and antifungal resistance profiles.

## 2. Materials and Methods

### 2.1. Patients, Strain Identification and Clinical Data

Between June and October 2023, clinical samples from eight patients admitted to São João Local Health Unit (ULSSJ), the biggest hospital in northern Portugal, were positive for *C. auris*. In five patients, *C. auris* was isolated in Sabouraud dextrose agar from routine surveillance screening swabs (axilla and groin) and urine. In three patients, *C. auris* was isolated from positive blood cultures. All isolates were identified by matrix-assisted laser desorption ionisation time-of-flight mass spectrometry (Vitek MS V3^®^) (bioMérieux, Craponne, France). The clinical data of the patients at the time of the first *C. auris* sample collection are summarised in Table 1. The Ethical competent authority approved this study—CES (Comissão de Ética para a Saúde, ULSSJ) and the Administration Council and Epidemiology Unit of ULSSJ—project code 67-21.

### 2.2. DNA Extraction

Genomic DNA was extracted following the instructions in the manual of the NZY Plant/Fungi gDNA Isolation kit, with some modifications. Yeast cells were disrupted in tubes containing zirconium beads and lysis buffer after ten cycles of 30 s each, at maximum velocity, using the Minilys homogeniser (Bertin Technologies, Montigny-le-Bretonneux, France). The Nanodrop (Thermo Fisher Scientific, Wilmington, NC, USA) and agarose electrophoresis assessed genomic DNA quantity and quality.

### 2.3. Genome Sequencing and Assembly

Yeast strains were sequenced from 1200 ng of genomic DNA. Library construction was performed using the Home-made Whole-Genome Sequencing prep kit (based on the Kapa HyperPrep kit). The integrity of the library fragment size was checked with the 4200 TapeStation System (Agilent Technologies, Santa Clara, CA, USA), and its concentration was measured using the Nanodrop ND-1000 spectrophotometer (Thermo Fisher Scientific, Wilmington, USA). The generated DNA fragments were sequenced in the Illumina Novaseq platfor (m, using 150 bp paired-end sequencing reads. After trimming the low-quality reads from output reads, the raw data quality was checked with the FastQC software (version 0.12.1) [16]. The trimmed sequences were assembled using CLC Genomics Workbench v.12.0.3 (https://www.qiagenbioinformatics.com/, accessed on 27 August 2025).

The high-quality sequencing reads were mapped against the reference genome of *C. auris* (Clade I) strain B8441, isolated from a patient’s blood in Pakistan. The trimmed sequence reads were used to perform a de novo assembly approach using an algorithm based on de Bruijn graphs. After the initial contig creation, the reads were mapped back to the contigs for assembly correction. The genome assembly quality was evaluated with QUAST v.5.1.0 [17].

### 2.4. InDel and Structural Variants Detection

After the mapping, a variant calling algorithm was applied to detect the variants that satisfy the requirements specified by the following filters: direction filtering = remove variants that do not have a statistically similar distribution in the forwards and reverse reads; minimum frequency = 20%; minimum quality = 25%; and minimum coverage = 50%. InDel and Structural Variants were detected to obtain the list of insertions and deletions in the samples based on the following criteria: minimum number of reads = 30; and *p*-value threshold = 0.05.

### 2.5. Phylogenomic Analysis

Phylogenomic analysis of clinical strains of *C. auris* was performed for accurate species identification. *C. auris* strains of the six prevalent clades (clades I–VI) were selected for phylogenetic reconstruction (Appendix A). The phylogenetic inference was carried out with the Reference sequence Alignment-based Phylogeny builder (REALPHY v.1.13) (https://realphy.unibas.ch/realphy/, accessed on 27 August 2025) [18], which uses input raw short sequence read data with read length parameter 150 (RL  =  150). The tree was constructed based on the maximum-likelihood principle, PhyML, optimised for speed in handling long sequence alignments. The general time-reversible (GTR) and gamma-distributed rate was used as a model of nucleotide evolution, as set by default. Constructed phylogeny was visualised and edited using the Interactive Tree of Life website (iTOL v.6) (https://itol.embl.de/, accessed on 27 August 2025) [19].

### 2.6. Antifungal Susceptibility Profile

Minimal inhibitory concentrations (MIC) for fluconazole (FLC), voriconazole (VRC), Posaconazole (PSC), isavuconazole (ISC), amphotericin B (AmB) and echinocandins, caspofungin (CSF), anidulafungin (ANF) and micafungin (MCF) were determined by microdilution assay according to the European Committee on Antimicrobial Susceptibility Testing E.Def.7.4 (EUCAST) and according to the M27-A3 protocol of the Clinical laboratory Standard Institute (CLSI) [20,21,22]. Since there are currently no established *C. auris*-specific susceptibility breakpoints, the strains were classified by using tentative breakpoints (TBP), according to CDC guidelines (https://www.cdc.gov/candida-auris/hcp/laboratories/antifungal-susceptibility-testing.html?CDC_AAref_Val=https://www.cdc.gov/fungal/candida-auris/c-auris-antifungal.html, accessed on 27 August 2025), epidemiological cutoff values (ECVs) defined by M27-M57S protocol [23] and the recently defined epidemiological cutoffs (ECOFF) determined by EUCAST [20].

### 2.7. Molecular Detection of Antifungal Resistance Gene Mutations

Several genes involved in antifungal resistance in *C. auris* were selected based on their associated mechanisms in drug resistance or tolerance (Table 2). The sequences of all genes analysed were extracted from the genome of *C. auris* B8441 (clade I) available in the NCBI database (https://www.ncbi.nlm.nih.gov/, accessed on 27 August 2025). Reference genes were aligned against the genome assemblies generated in this study using BLAST+ v.2.11.0 [24] to identify the corresponding gene regions. Variant calling for single-nucleotide polymorphisms (SNPs) and small insertions/deletions (indels) was performed using FreeBayes v1.3.10 with default parameters. Variants were annotated and translated into amino acid substitutions using SnpEff v5.2 [25]. All mutations were further confirmed by visually inspecting the alignment of the reads in Integrative Genomics Viewer v2.1.2 [26]. Moreover, antifungal resistance point mutations in the genes were assessed and confirmed using the Solu online platform (Solu Healthcare Inc., Helsinki, Finland, https://platform.solu.bio/, accessed on 27 August 2025). Alignments of genes and corresponding amino acid sequences were generated with Clustal Omega v1.2.4 (https://www.ebi.ac.uk/jdispatcher/msa/clustalo, accessed on 27 August 2025).

**Table 1 jof-11-00716-t001:** Demographic and clinical characteristics of patients with *C. auris* positive cultures.

	Candidemia			Colonisation				
	SCO 267	SCO 276	SCO 279	SCO248	SCO 275	SCO 240	SCO242	SCO266
Sex/age	Male/66	Male/67	Female/58	Female/64	Male/51	Male/59	Male/77	Male/63
Sample	blood culture	blood culture	blood culture	CVC	CVC	rectal swab	urine	axillary and groin swab
Hospital ward	ICU	surgery	medicine	infectiology	orthopaedics	surgery	surgery	surgery
Days of hospitalisation until the first isolation	91	106	110	22	132	34	15	79
ICU before isolation	yes	yes	no	yes	no	yes	no	no
Antifungal treatment	micafungin	Caspofungin + amphotericin B	caspofungin	no	no	no	no	no
Underlying disease	Oesophageal neoplasia, dyslipidemia	Gastric adenocarcinoma, peritonitis, HTN, dyslipidemia	Severe gonarthrosis, infection after total knee arthroplasty, DM2, HTN, dyslipidemia	hepatic abscess by *Clostridium difficile*, DM2, HTN, dyslipidemia	chronic osteomyelitis	Colon neoplasia, faecal peritonitis	Pancreatic adenocarcinoma DM2, HTN	Biliary tract neoplasia, acute cholangitis with hepatic abscess
Outcome	deceased	deceased	alive	alive	alive	alive	alive	deceased

DM2: diabetes mellitus type 2; HTN: hypertension; CVC: central venous catheter.

**Table 2 jof-11-00716-t002:** Genes analysed for antifungal resistance in *Candida auris*, grouped by functional category. The table lists the gene categories, individual genes, and their associated mechanisms or functions in drug resistance or tolerance.

Category	Genes	Mechanism/Function	References
Drug targets and ergosterol biosynthesis (Azoles/Polyenes)	*ERG11*	Azole target; mutations reduce drug binding	[27,28]
	*ERG2*, *ERG3*, *ERG4*, *ERG5*, *ERG6*, *ERG10*, *ERG25*	Enzymes in ergosterol pathway; alterations affect membrane composition and polyene/azole susceptibility	
Efflux pumps (ABC/MFS transporters)	*CDR1*, *CDR2*, *CDR4*, *MDR1*, *SNQ2*	Drug transporters that expel antifungals; overexpression or mutations increase azole resistance	[29,30]
Transcription factors	*TAC1B*, *UPC2*, *MRR1*, *CRZ1*	Regulate expression of efflux pumps or ergosterol pathway; *CRZ1* modulates stress response and tolerance	[29,31]
Epigenetic regulators	*GCN5*, *SET1*, *SET2*, *RTT109*, *DOT1*	chromatin remodelling and antifungal stress response gene expression	[32]
Echinocandin target/1,3-β-glucan synthase	*FKS1*, *FKS2*	Subunits of 1,3-β-glucan synthase; hotspot mutations confer echinocandin resistance	[33]
Stress response and antifungal tolerance	*HSP90*, *CNA1*	Chaperone and calcineurin subunit; modulate stress response and tolerance to echinocandins/azoles	[34]
Other tolerance/stress and metabolism	*CIS2*	Cystathionine γ-lyase; involved in sulfur metabolism, redox homeostasis, and stress response	[35]
Pheromone/export and signalling	*STE6*	ABC transporter for a-factor pheromone export; may influence mating and signalling	[36]

## 3. Results

### 3.1. Case Patient Description and Timeline

Clinical characteristics of the three candidemia and five asymptomatic carriage patients with *C. auris* are detailed in Table 1.

Three colonisation sites were identified: in the skin (2 patients, isolates SCO240 and SCO266), in the central venous catheter (2 patients, isolates SCO248 and SCO275), and in urine (1 patient, SCO242). The patient median age was 63.1 (range: 51–77); two were females, and six were males. The patients were hospitalised between May and October 2023, with four patients simultaneously admitted to the same ward (surgery, isolates SCO276, SCO240, SCO242, SCO266) (Figure 1).

SCO240 and SCO242 isolates were isolated from patients sharing the same room, and SCO266 and SCO276 were recovered from patients in close rooms. The median time from admission to the first *C. auris* isolation was 106 and 34 days for candidemia and colonised patients, respectively; however, isolation times vary from 15 to 132 days. Three patients developed *C. auris* candidemia (isolates SCO267, SCO276 and SCO279); however, no information regarding colonisation sites or early detection was obtained before the onset of the invasive infection. Only candidemia patients received antifungal therapy, mainly with echinocandins. Three candidemia patients died (infected with SCO266, SCO267 and SCO276 strains), although death was not exclusively attributed to *C. auris* infection. All the patients had no recent travel history, nor were they transferred from other hospitals or healthcare facilities. No environmental samples were obtained, nor were contacts outside the hospital, such as family members, swabbed.

### 3.2. Genome Statistics and Phylogenomic Analysis

The genome assemblies of the eight *C. auris* strains showed an average length of 12.2 Mb, a contig count of 7, GC content of 45.3%, a scaffold N50 value of 2.3 bp, and a coverage depth for quality-trimmed reads ranging from 149-fold to 247-fold (Table 3). The accession numbers for the assemblies generated in this study are presented in Table 3. *C. auris* clade I, commonly called the South Asian, comprises isolates from cases in India, Pakistan, the United Arab Emirates, Lebanon, Italy, China, and Brazil. Phylogenetic analysis using whole-genome sequencing revealed that all resistant *C. auris* strains analysed in this study (SCO strains) belong to Clade I (Figure 2).

### 3.3. Genetic Variants Detection

The genetic variant analysis of *C. auris* strains revealed the presence of different mutation types, including single-nucleotide variants (SNVs), deletions, insertions, replacements, and multi-nucleotide variants (MNVs) (Appendix A). The total number of SNVs was consistent across the strains, ranging from 60,619 (SCO 267) to 61,659 (SCO 275), varying by approximately 1000 variants. Regarding deletions, strain SCO 275 showed the highest count (2418), while strain SCO 276 had the lowest (2214). As for insertions, the numbers ranged from 3101 (SCO 276) to 3674 (SCO 275), with SCO 275 also showing the highest number of such variants. The strains displayed more minor variations for replacements, with SCO 275 having the highest number (337) and SCO 276 having the lowest (308). Finally, the MNVs, referring to variants involving multiple nucleotides, also exhibited similar numbers across strains, with SCO 275 showing the highest (3224) and SCO 267 being the lowest (3130) (Appendix A).

### 3.4. Antifungal Susceptibility

Antifungal MIC values, determined according to EUCAST, susceptibility profiles, and phenotypes of the *C. auris* isolates from colonised patients and patients with candidemia are presented in Table 4. MIC values and susceptibility profiles determined according CLSI protocol are detailed in Appendix A [21,22]. All isolates showed high MIC values to fluconazole and amphotericin B, categorised as a resistant phenotype according to EUCAST ECOFFs, being one isolate (SCO242) classified as increased exposure (IE) (Table 4) [20,37]. Most of the isolates showed a resistant phenotype to anidulafungin and micafungin, and all isolates were resistant to caspofungin.

### 3.5. Analysis of Resistance-Associated Mutations

We identified several mutations in antifungal resistance–associated genes across our *C. auris* isolates (Figure 3, Table 5). In the ergosterol biosynthesis pathway, the *ERG11* gene carried the missense mutation Y132F (c.396A>T), a well-characterised mutation conferring resistance to azoles [38,39]. In *ERG4*, we detected the substitution M192I (c.576G>T), which may affect sterol composition and modulate azole susceptibility. Among efflux-related genes, the *CDR1* gene contained the substitution E709D (c.2126A>T), while its transcriptional regulator *TAC1b* harboured the A583S (c.861C>A) mutation, potentially impacting efflux activity. In the ABC transporter *SNQ2*, two additional variants were identified: K52N (c.72C>T) and E1464K (c.4306T>A), both previously associated with resistant isolates and possibly contributing to efflux-mediated azole tolerance [40].

Although its contribution to antifungal resistance in *C. auris* is negligible, a novel substitution in the transcription factor *CRZ1* (S237Y, c.710C>A) was detected [32].

Concerning to recently epigenetic regulators involved in antifungal resistance [32], an in-frame deletion of six nucleotides in the *GCN5* gene (c.439_444del) was detected in five isolates (SCO 240, SCO 248, SCO 266, SCO 275 and SCO 279) comparative to the B8441 reference genome, resulting in the deletion of two amino acids (p.(Glu134_Asn135del)) (Table 5). This variant was confirmed by inspection of the read alignments against the assemblies. The deletion lies within the N-terminal low-complexity region of Gcn5, whose functional significance is unknown. Other epigenetic regulators, *STE1*, *SET2*, *RTT109* and *DOT1* were also screened for genomic mutations, but no variants were found.

Similarly, no variants were detected in *FKS1*, *FKS2*, *CDR2*, *CDR4*, *ERG2*, *ERG3*, *ERG5*, *ERG6*, *ERG 10*, *ERG 25*, *MDR1*, *MRR1*, *HSP90*, *CNA1*, *UPC2*, and *CIS2* genes in the set of *C. auris* clinical isolates, suggesting that it is unlikely that these genes play a role in the resistance phenotypes observed in these isolates. In the *STE6* gene, a synonymous substitution (c.2157G>T) was identified, causing no alteration in the amino acid sequence and therefore, with no protein functional impact.

## 4. Discussion

The interaction between *C. auris* and the skin microbiota is complex, influencing both the persistence of the pathogen on the host and its potential for transmission [3]. Colonisation often precedes infection, with the skin microbiota serving as a critical reservoir for the yeast [41,42]. Understanding these dynamics is crucial for developing effective infection control and prevention strategies. Whole-genome sequencing allows, for instance, detailed characterisation of *C. auris* strains from colonised patients, providing insights into their genetic diversity, antifungal resistance profiles, and potential transmission pathways. Epidemiological studies have reported colonisation rates ranging from 5% to 70%, depending on the geographic location, patient population, and local infection control practices [41,42,43].

We report for the first time in Portugal, eight cases of candidemia and colonisation with *C. auris* in 2023, with potential epidemiological linkage. Most patients enrolled in this study had multiple health conditions, such as diabetes mellitus or neoplasia, yet only one patient had a prior ICU admission. The median from hospital admission to onset of candidemia was 106 days, longer than the time reported by other studies [44,45]. This is consistent with the notion that *C. auris* infections frequently occur several weeks after hospital admission, particularly among patients with prolonged hospital stays, multiple admissions, or those requiring intensive care [1,46]. These risk factors for *C. auris* infections are closely tied to healthcare settings and involve critical illness, invasive devices, antimicrobial use, and immunosuppression [1,47,48]. Effective infection control practices, early diagnosis, and antifungal stewardship are essential to mitigate risk.

Whole-genome sequencing revealed that all isolates belonged to Clade I, with single nucleotide variant (SNV counts ranging from 60,619 to 61,058. While crude SNP counts alone are insufficient to infer genetic relatedness accurately, it is plausible that all strains of *C. auris*, which belong to Clade I, were hospital-acquired, potentially originating from patients admitted simultaneously and sharing rooms. Two candidemia patients (SCO267 and SCO276) and one colonised patient (SCO266) died, most probably due to their complex health state and other concomitant pathologies. Future analyses using MLST or core genome phylogenomic approaches would be required to confirm these isolates’ genetic relationships and transmission pathways.

Antifungal susceptibility testing revealed a concerning multidrug-resistance profile. All isolates showed high MIC values to fluconazole and amphotericin B according to EUCAST ECOFFS [20,37], consistent with the high rates of azole resistance reported in Clade I isolates [1,28,35]. Resistance to fluconazole is a hallmark of Clade I isolates, with studies consistently reporting resistance rates exceeding 90% [47]. For instance, in a large-scale survey of *C. auris* isolates from India, more than 90% of Clade I isolates were resistant to fluconazole, while up to 5% were resistant to echinocandins [1,49]. In our study, voriconazole and posaconazole exhibited slightly better activity against *C. auris* isolates than fluconazole, although cross-resistance limits their therapeutic utility [1]. MIC values for isavuconazole ranged from 0.015 to 2 µg/mL. Although no ECVs or breakpoints are established for isavuconazole, MIC values vary significantly across studies and clades, including Clade I [1,50]. For instance, a global surveillance study of *C. auris* isolates from Clade I display MICs ≥4 µg/mL, indicating reduced susceptibility [50]. As frequently happens when using azole therapeutics, cross-resistance to isavuconazole and fluconazole is common in Clade I isolates due to similar target-binding sites in the lanosterol 14α-demethylase enzyme, limiting the utility of isavuconazole as a treatment option for fluconazole-resistant isolates [47,50,51]. Clinical use of isavuconazole for *C. auris* infections should be guided by antifungal susceptibility testing, and it is typically considered only when other triazoles and echinocandins are ineffective or contraindicated [51,52].

Resistance to azoles arises from the overexpression of efflux pumps, ATP-binding cassette (ABC) transporters, such as *CDR1* and *MDR1*, and mutations in the *ERG11* gene, the target of azoles [31]. Mutations in the *ERG11* gene, such as Y132F and K143R, can reduce drug binding and are commonly suggested as responsible for the cross-resistance mechanisms shared between azole drugs [53,54]. The *ERG11* mutation Y132F was detected in our cohort, consistent with the observed fluconazole resistance. The E709D mutation in the *CDR1* gene was also identified. However, its role in antifungal resistance remains unclear and may act in combination with other resistance-associated mutations, such as A583S in the *TAC1B* gene [55]. Mutations in *CDR1* and *TAC1b* may further contribute to antifungal resistance by enhancing efflux activity [56]. However, further research is necessary to elucidate the functional impact and specific roles of E709D and A583S mutations in mediating antifungal resistance in *C. auris.* Two variants in the *SNQ2* ABC transporter (K52N, E1464K) were detected. While their functional impact is poorly characterised, upregulation of *SNQ2* has been associated with multidrug resistance in clinical isolates, suggesting a potential role in modulating antifungal susceptibility. These findings highlight the contribution of efflux systems to antifungal tolerance in *C. auris*, demanding further functional validation.

Although less common, amphotericin B resistance involves alterations in the ergosterol biosynthesis pathway, which may impact membrane integrity; however, it can be balanced by lipid metabolism and membrane structure modifications [28]. Resistance to amphotericin B is variably reported in Clade I isolates, with rates ranging from 10% to 30% [47]. Loss-of-function mutations in the *ERG6* and *ERG3* genes, involved in ergosterol biosynthesis, have been associated with amphotericin B resistance [57,58]. In this study, all *C. auris* isolates displayed a resistance profile to amphotericin B, with one considered susceptible with increased exposure according to EUCAST [20,37]. However, no mutations were detected in *ERG6* and *ERG3*. Instead, a missense substitution (M192I) was identified in *ERG4*, a downstream enzyme in the ergosterol pathway. Such changes may affect sterol compositions, potentially contributing to amphotericin B tolerance. In *S. cerevisiae*, *ERG4* modifications have been shown to influence polyene sensitivity [59], thus corroborating that downstream sterol biosynthetic enzymes can modulate antifungal susceptibility. Nevertheless, no evidence links the *ERG4* M192I variant to antifungal resistance in *C. auris*. Future studies are needed to determine whether the *ERG4* M192I mutation modifies membrane sterol composition and consequently affects drug tolerance.

*C. auris* Clade I isolates are usually considered susceptible to echinocandins, including caspofungin, micafungin, and anidulafungin [55,60]. However, emerging reports have described the development of echinocandin resistance in Clade I strains [60]. In this study, most isolates exhibited a resistant phenotype for anidulafungin (7 out of 8) and micafungin (6 out of 8), and all isolates were resistant to caspofungin. However, when MIC values were determined and interpreted according to the CLSI protocol [21], a wild-type phenotype to anidulafin and micafungin was observed. Typically, echinocandin resistance is usually associated with mutations in the *FKS1* and *FKS2* genes, encoding the β-1,3-glucan synthase, often resulting in a fitness cost due to cell wall synthesis disruption [49,61]; nevertheless, no mutation in these genes was detected in our isolates.

Recently, the role of epigenetic regulators in *C. auris* antifungal resistance was comprehensively described [36]. Notably, the gene encoding the acetyltransferase catalytic subunit *GCN5*, responsible for histone H3 acetylation, was shown to have a crucial role in the transcription of genes involved in ergosterol biosynthesis, drug efflux, cell wall integrity and echinocandin resistance through the calcineurin signalling pathway. Therefore, Gcn5 seems to be a master regulator for *C. auris* drug resistance, associated with azole and polyene tolerance, and echinocandin resistance [32]. Interestingly, in the set of *C. auris* isolates, five of them displayed a variant of the *GCN5* gene (c.439_444del), resulting in the in-frame deletion of two amino acids (Glu134_Asn135del), whose functional impact is unknown. Although the deletion we report is partial and located outside the annotated catalytic HAT domain, its recurrent presence in multiple clinical isolates raises the possibility of a modulatory effect on Gcn5 function. Definitive conclusions will require targeted functional assays, and we report this variant as an intriguing candidate for further investigation of GCN5-mediated pathways in antifungal tolerance. Besides *GCN5*, other genes such as *SET1*, *SET2*, *RTT109* and *DOT1* were recently associated with azole and echinocandin tolerance [36]. In the set of *C. auris* isolates, no mutation was detected in these genes.

We also identify a novel missense mutation in the calcineurin pathway target *CRZ1* (S237Y). *CRZ1* is a transcription factor involved in stress response and cell wall integrity, whose role in drug resistance is still controversial among *Candida* species [34,62,63]. In *C. auris*, growth under caspofungin exposure increases the expression of the CRZ1 gene in a Gcn5-dependent manner. Furthermore, the *crz1Δ* mutants displayed the same sensitivity to caspofungin as the wild-type strain, dissociating its role in echinocandin resistance [32].

In addition to the canonical resistance-associated genes, we investigated *CIS2* and *STE6*, which have been implicated in experimental or clinical studies. Recently, a substitution in CIS2 (A27T), encoding a γ-glutamyltranspeptidase involved in xenobiotic detoxification, was linked to increased echinocandin MICs in experimentally evolved *C. auris* strains [35]. Another variant (K74E) was reported in susceptible and resistant isolates, suggesting some *CIS2* polymorphisms may not be under antifungal selective pressure. In our isolates, no nonsynonymous variants were detected in *CIS2*. Regarding *STE6*, an a-pheromone ABC transporter reported to modulate antimycotic responses [36], we identified only a synonymous substitution (c.2157G>T), which is unlikely to impact protein function. Nevertheless, given the potential involvement of these genes in antifungal tolerance and adaptation, continued surveillance and functional studies are warranted.

This study provides several insights regarding *C. auris* infections. However, several limitations warrant consideration. First, the small sample size restricts the strength of the evidence for predicting risk factors associated with *C. auris* candidemia. Second, the lack of comprehensive surveillance cultures, including samples from various body sites, the hospital environment, and healthcare staff, limits the understanding of transmission patterns among patients. Also, the study’s retrospective nature precluded the collection of a complete exposure history. Despite the limitations, our findings provide important insights into the genetic determinants of antifungal resistance and potential transmission pathways of *C. auris* in Portuguese healthcare settings. Understanding these mechanisms is crucial to developing novel therapeutic approaches and targeting infection control strategies to guide the formulation of more robust hygiene practices, environmental cleaning protocols, and patient management strategies to reduce the spread of *C. auris*. This will guide future research into the molecular basis of resistance and persistence in this emerging pathogen.

## Figures and Tables

**Figure 1 jof-11-00716-f001:**
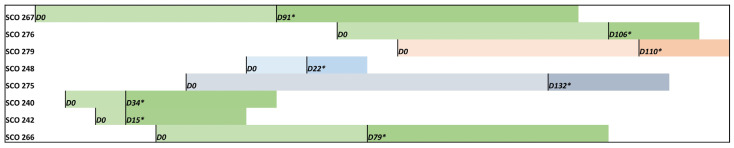
Timeline representing hospitalisation days (*D0*—admission) and the first *C. auris* positive detection (*) from colonised and candidemia patients. Green colour refers to patients admitted to the same hospital ward (surgery), grey to orthopaedics, light blue to infectiology and salmon to medicine.

**Figure 2 jof-11-00716-f002:**
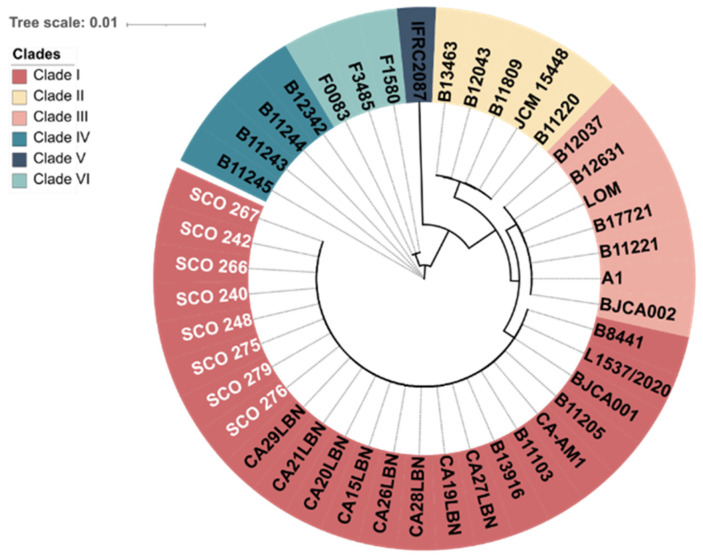
Phylogenetic tree showing the genetic relationships among isolates from six distinct clades. Resistant *Candida auris* isolates from this study are highlighted in white. The scale bar indicates the evolutionary distance.

**Figure 3 jof-11-00716-f003:**
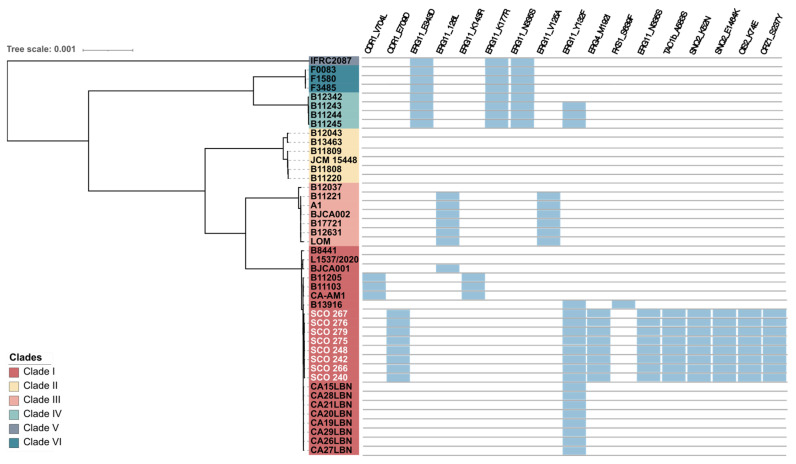
Summary of the putative molecular mechanisms underlying antifungal resistance (blue boxes) found in *C. auris* isolates belonging to the six clades identified. The eight isolates from the present study are highlighted in white. The scale bar indicates the evolutionary distance.

**Table 3 jof-11-00716-t003:** Genomic features of eight clinical strains of *Candida auris* from this study.

General Features	Strains							
	SCO 240	SCO 242	SCO 248	SCO 266	SCO 267	SCO 275	SCO 276	SCO 279
Total length (bp)	12,252,649	12,252,724	12,252,809	12,252,617	12,253,318	12,252,600	12,252,471	12,252,602
Contigs	7	7	7	7	7	7	7	7
Largest contig (bp)	3,148,704	3,148,755	3,148,719	3,148,682	3,148,861	3,148,688	3,148,687	3,148,708
GC (%)	45.34	45.34	45.34	45.34	45.33	45.34	45.34	45.34
N50 (bp)	2,337,416	2,337,416	2,337,434	2,337,386	2,337,558	2,337,379	2,337,346	2,337,360
N75 (bp)	1,318,631	1,318,648	1,318,659	1,318,666	1,318,682	1,318,617	1,318,615	1,318,621
L50 (bp)	3	3	3	3	3	3	3	3
L75 (bp)	4	4	4	4	4	4	4	4
Coverage	225×	214×	212×	223×	149×	230×	247×	214×
GenBank Accession Number	CP163311-CP163317	CP163318-CP163324	CP163325-CP163331	CP163332-CP163338	CP163339-CP163345	CP163346-CP163352	CP163353-CP163359	CP163360-CP163366

**Table 4 jof-11-00716-t004:** Antifungal minimal inhibitory concentration (MIC) and susceptibility phenotype of *C. auris* isolates according to EUCAST guidelines for antifungal susceptibility testing.

	FLC		VRC		PSC		ISC		ANF		MCF		CSF		AmB	
**Clinical isolate**	MIC	Phen	MIC	Phen	MIC	Phen	MIC	Phen	MIC	Phen	MIC	Phen	MIC–	Phen**#**	MIC	Phen
**SCO 240**	>64	ND	0.125	ND	0.03	ND	0.03	ND	0.5	**R**	0.125	S	0.5	**R**	4	**R**
**SCO 242**	>64	ND	0.125	ND	0.03	ND	0.03	ND	0.5	**R**	0.25	S	>4	**R**	1	IE
**SCO 248**	>64	ND	0.06	ND	1	ND	0.0075	ND	2	**R**	2	**R**	>4	**R**	4	**R**
**SCO 266**	>64	ND	0.5	ND	0.03	ND	0.06	ND	2	**R**	1	**R**	>4	**R**	4	**R**
**SCO 267**	>64	ND	0.5	ND	0.03	ND	0.06	ND	2	**R**	1	**R**	>4	**R**	4	**R**
**SCO 275**	>64	ND	0.06	ND	1	ND	0.015	ND	4	**R**	4	**R**	>4	**R**	4	**R**
**SCO 276**	>64	ND	0.125	ND	0.015	ND	0.03	ND	0.25	S	0.5	**R**	>4	**R**	4	**R**
**SCO 279**	>64	ND	0.125	ND	2	ND	0.0075	ND	4	**R**	4	**R**	1	R	4	**R**

FLC—Fluconazole; VRC—Voriconzole; PSC—Posaconazole; ISC—Isavuconazole; ANF—Anidulafungin; MCF—Micafungin; CSF—Caspofungin; AmB—Amphotericin B. Phenotype (Phen) at 24 h incubation, based on EUCAST epidemiological cutoff (ECOFF). ND—not defined; S—Susceptible; IE—Increased exposure; R—Resistant.

**Table 5 jof-11-00716-t005:** List of variants reported in *Candida auris* isolates in Portugal. Newly reported variants are highlighted in bold.

Gene	Change	Mutation	Variant	Isolates of This Study
*CIS2*	c.220A>G p.Lys74Glu	K74E	Missense variant	all
*ERG4*	c.576G>T p.Met192Ile	M192I	Missense variant	all
*SNQ2*	c.72C>T (p.Lys52Asp)	K52N	Missense variant	all
	c.4306T>A (p.Glu1464Lys)	E1464K	Missense variant	all
*ERG11*	c.396A>T (p.Tyr132Phe)	Y132F	Missense variant	all
*CDR1*	c.2126A>T (p.Glu709Asp)	E709D	Missense variant	all
*TAC1b*	c.861C>A (p.Ala709Ser)	A583S	Missense variant	all
*STE6*	c.2157G>T	-	Synonymous variant	all
** *CRZ1* **	**c.710C>A (p.Ser237Tyr)**	**S237Y**	**Missense variant**	**all**
** *GCN5* **	**c.439_444del**	**p.(Glu134_Asn135del)**	**In-frame deletion**	**SCO 240, SCO 248, SCO 266, SCO 275, SCO 279**

## Data Availability

The original contributions presented in this study are included in the article/Appendix A. Further inquiries can be directed to the corresponding author.

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
