# Peer review of "First Report of *Candida auris* Candidemia in Portugal: Genomic Characterisation and Antifungal Resistance-Associated Genes Analysis"

_jof, 2025, doi:10.3390/jof11100716_

Round 1

Reviewer 1 Report

The title is somehow misleading, since there was a report of the first colonization in Portugal in 2022. Consider revising the title; e.g. “Genomic characterization of C. auris candidemia in Portugal reveals novel FKS1 mutation and…”. Alternatively, remove the word “colonization”, if this is the first case of candidemia in Portugal.

There is some issues and/or information lacking from the result section “Genetic variants detection”. Which reference genome was used for alignment and analysis? In figure 4, strain B11220 is used as reference. However, this is a clade II strain. The reference strain (e.g. strain on Candida Genome Database) for C. auris genomic analysis is strain B8441 of Clade I (same clade as isolates of this study), which should be used here as well for analysis. Please also discuss synonymous vs nonsynonymous variants. Also, compare variants between strains of this study, and with a closely related strain as reference (like B8441, or closer), to identify variants of interest in genes other than those investigated in the context of drug resistance.

The reported FKS1 mutation does not match the reference sequence. In the officially annotated sequence of FKS1 of C. auris, available here: http://www.candidagenome.org/cgi-bin/locus.pl?locus=B9J08_000964 The nucleotide at position 4393 is a Thymine (T), and the codon translates a Tryptophan (W), unlike the reported Adenine (A) translating an Isoleucine (I). This major issue might be the result of the use of a wrong reference sequence (see second comment), and the reported mutation might solely be a inter-clade substitution! In addition to being wrongly annotated, the FKS1 hotspot 1, is very far away from the reported mutation. Hotspot 1 spans nucleotides 1903-1929, nowhere near the reported “hotspot 1”. In fact, the sentence “In the gene encoding the β-1,3-glucan synthase FKS1 hotspot1 (HS1), a novel nucleotide 242 substitution at position A4393C resulted in the amino acid substitution I1465L (Figure 4).” is wrong, as “hotspot 1” is not a gene, it’s a region in the gene. I suggest revision of the whole genome analysis data by a genomic analysis expert.

Often, there is info missing regarding analyses and software in the methods section. For example, the variant calling algorithm is not described in detail, nor referenced. Which algoritm and program were used? Or in the case of “All 160 mutations were confirmed by visually inspecting the alignment of the reads to the corresponding assemblies.” How were they visually inspected? Using IGV on the alignment? Please provide more detailed methods.

The use of crude amount of SNPs as indicator of relatedness (line 318) is not okay. Please use reference methods such as multi-locus sequence typing (e.g. https://journals.asm.org/doi/full/10.1128/mbio.02971-19) or phylogenomics to type the strains.

Line 71-76; following study might be a better/extra reference for that statement: https://doi.org/10.1038/s41564-024-01854-z

Table 1 is too big and out of proportion.  

Line 193: “although death was not exclusively attributed to C. auris infection.” Could you report cause of death?

There is confusion regarding CLSI vs EUCAST. Which method was used? Why are both mentioned?

How were the genes chosen that were investigated in “Analysis of resistance-associated mutations”?

Table 3 is confusing and an unusual way of presenting this sort of data. Consider choosing another way of presenting it. In addition, please provide the susceptibility profiles (OD Measurements in function of drug concentration), a supplementary material. I am especially interested in the profile for caspofungin, as the eagle effect is known to distort MIC interpretations. With a strain being resistant to caspofungin and not micafungin or anidulafungin, supra MIC growth (tolerance) or the eagle effect might play a role in strain 275.  

What is the relevance of the start of the discussion (line 292-300)? Please remove information that is irrelevant to the manuscript.

Author Response

Point-by-point response to Reviewer 1

Comment 1: The title is somehow misleading, since there was a report of the first colonization in Portugal in 2022. Consider revising the title; e.g. “Genomic characterization of C. auris candidemia in Portugal reveals novel FKS1 mutation and…”. Alternatively, remove the word “colonization”, if this is the first case of candidemia in Portugal.

Response 1: Thank you very much for your suggestion. Considering the novel findings that resulted from genome blast to the Clade I reference strain (B8441), as suggested by the Reviewer, the authors changed the title to “First report of Candida auris candidemia in Portugal: Genomic characterisation and antifungal resistance-associated genes analysis”. The word colonization has been removed.

Comment 2: There is some issues and/or information lacking from the result section “Genetic variants detection”. Which reference genome was used for alignment and analysis? In Figure 4, strain B11220 is used as a reference. However, this is a clade II strain. The reference strain (e.g., on Candida Genome Database) for C. auris genomic analysis is strain B8441 of Clade I (same clade as isolates of this study), which should also be used for analysis. Please also discuss synonymous vs nonsynonymous variants. Also, compare variants between strains of this study, and with a closely related strain as reference (like B8441, or closer), to identify variants of interest in genes other than those investigated in the context of drug resistance.

Response 2: Thank you for your comment. When preparing the manuscript, the reference strain B8441 was not yet available in GenBank and therefore could not be included in our analysis. The genome of this strain was annotated on 22 April 2024 and only released in August 2024, by which point the manuscript had already been completed. We acknowledge, however, that we should have verified whether additional strains (including the reference strain) had become available during the review process. In line with the reviewer’s suggestion, we have included strain B8441 in the analysis and provided a table detailing the variants identified in our strains (Table 5, Figures 2 and 3, Supplementary Material Table S1).

Comment 3: The reported FKS1 mutation does not match the reference sequence. In the officially annotated sequence of FKS1 of C. auris, available here: http://www.candidagenome.org/cgi-bin/locus.pl?locus=B9J08_000964, the nucleotide at position 4393 is a Thymine (T), and the codon translates a Tryptophan (W), unlike the reported Adenine (A) translating an Isoleucine (I). This major issue might be the result of the use of a wrong reference sequence (see second comment), and the reported mutation might solely be a inter-clade substitution! In addition to being wrongly annotated, the FKS1 hotspot 1, is very far away from the reported mutation. Hotspot 1 spans nucleotides 1903-1929, nowhere near the reported “hotspot 1”. In fact, the sentence “In the gene encoding the β-1,3-glucan synthase FKS1 hotspot1 (HS1), a novel nucleotide 242 substitution at position A4393C resulted in the amino acid substitution I1465L (Figure 4).” is wrong, as “hotspot 1” is not a gene, it’s a region in the gene. I suggest revision of the whole genome analysis data by a genomic analysis expert.

Response 3: Thank you for your comment. We understand the reviewer’s concern regarding using an incorrect reference sequence. As mentioned in our previous response, the reference strain B8441 was unavailable during manuscript preparation. However, as suggested, we have included the Clade I reference strain (B8441) in our analysis. Upon performing the FKS1 gene alignments, we found that the previously reported FKS1 mutation does not match the reference sequence. Consequently, we removed Figure 4 and replaced it with a table summarising the nucleotide and protein variants identified in our strains (Table 5).

Comment 4: Often, there is info missing regarding analyses and software in the methods section. For example, the variant calling algorithm is not described in detail, nor referenced. Which algoritm and program were used? Or in the case of “All 160 mutations were confirmed by visually inspecting the alignment of the reads to the corresponding assemblies.” How were they visually inspected? Using IGV on the alignment? Please provide more detailed methods.

Response 4: Thank you for your comment. We have clarified and expanded the Methods section to provide detailed information on detecting and confirming antifungal resistance gene mutations. Specifically, we now describe the variant calling algorithm used, the annotation and translation of variants into amino acid substitutions, and the procedure for visual confirmation of all mutations using Integrative Genomics Viewer (lines 170-183).

Comment 5: The use of crude amount of SNPs as indicator of relatedness (line 318) is not okay. Please use reference methods such as multi-locus sequence typing (e.g. https://journals.asm.org/doi/full/10.1128/mbio.02971-19) or phylogenomics to type the strains.

Response 5: We thank the reviewer for this valuable comment. We agree that crude SNP counts alone cannot infer genetic relatedness among strains. In the revised manuscript, we have clarified that the reported SNP differences cannot be used as definitive evidence of relatedness. We also highlight that future analyses using multi-locus sequence typing (MLST) or core genome phylogenomics would be necessary to accurately assess these isolates' genetic relationships and potential transmission pathways (lines 373-380).

Comment 6: Line 71-76; following study might be a better/extra reference for that statement: https://doi.org/10.1038/s41564-024-01854-z

Response 6:  The reference was added (line 88).

Comment 7: Table 1 is too big and out of proportion.

Response 7: In the submitted manuscript version, Table 1 is now Table 2 and has been redimensioned.

Comment 8: Line 193: “although death was not exclusively attributed to C. auris infection.” Could you report cause of death?

Response 8: In fact, it would be interesting to point to sepsis as the cause of death; however, these patients, besides candidemia, displayed a complex state of disease, including neoplasia. Therefore, it would be incorrect to attribute candidemia as the cause of death.

Comment 9: There is confusion regarding CLSI vs EUCAST. Which method was used? Why are both mentioned?

Response 9: MICs were determined for each isolate following EUCAST (Table 4) and CLSI (Table S3 - Supplementary material) protocols and breakpoints. Findings were separated, and we opted to describe MICs and susceptibility profiles of our isolates according to EUCAST guidelines in the main text (Table 4). At the same time, results from the CLSI protocol can be consulted in the supplementary material (Table S3).

Comment 10: How were the genes chosen that were investigated in “Analysis of resistance-associated mutations”?

Response 10: The genes are summarised in Table 1. The drug resistance-associated genes were chosen according to previous reports and are involved in known fungal resistance mechanisms to antifungal drugs: the ergosterol biosynthesis pathway, efflux pumps and their transcription factors, echinocandins targets (FKS genes), epigenetic regulators, and others.

Comment 11: Table 3 is confusing and an unusual way of presenting this sort of data. Consider choosing another way of presenting it. In addition, please provide the susceptibility profiles (OD Measurements in function of drug concentration), a supplementary material. I am especially interested in the profile for caspofungin, as the eagle effect is known to distort MIC interpretations. With a strain being resistant to caspofungin and not micafungin or anidulafungin, supra MIC growth (tolerance) or the eagle effect might play a role in strain 275.  

Response 11: Table 3 was reformulated, simplified, and replaced by Table 4. MIC values were determined according to the European Committee on Antimicrobial Susceptibility Testing E.Def.7.4 (EUCAST) protocol. Despite high MIC levels to caspofungin, no eagle effect was observed.

Comment 12: What is the relevance of the start of the discussion (lines 292-300)? Please remove information that is irrelevant to the manuscript.

Response 12: Thank you very much for your comment. The discussion was changed, and inadequate information was removed.

Reviewer 2 Report

Miranda et al., present a timely and relevant investigation into the emergence of Candida auris in Portugal, combining clinical data with genomic and antifungal susceptibility analyses. Despite its relevance, the manuscript must be improved for publication in JoF

Major concerns

1- The identification of a novel missense mutation (I1465L) in the FKS1 hotspot region is a potentially significant finding, given its location in a conserved domain of the β-1,3-glucan synthase. However, the manuscript does not provide functional evidence to support its role in echinocandin resistance.  Even if experimental validation is beyond the scope of this study, a more thorough discussion of the mutation’s potential impact and its novelty in the context of existing literature would strengthen the manuscript.

2- The manuscript presents MIC data interpreted using multiple standards (CLSI, CDC tentative breakpoints, EUCAST ECOFFs), which may confuse readers. Some isolates are described as “non-wild-type” under EUCAST but “susceptible” under CLSI, without a clear rationale for prioritizing one classification over another.

3- The study includes only eight isolates from a single hospital, which limits the generalizability of the findings. While the novelty of the report is appreciated, broader conclusions about transmission, resistance patterns, or clade behavior should be framed cautiously.

Minor concerns

4- Minor grammatical and stylistic issues are present throughout the manuscript.  For example, Introduction: Consider rephrasing “pressing need for global surveillance” to “urgent need for enhanced global surveillance.”

    A thorough proofreading pass is recommended to improve flow and readability.

5- Table 3 (MICs and Susceptibility Profiles): The table is dense and includes multiple interpretive standards (CLSI, ECOFF, TBP). A simplified version or a supplementary table comparing interpretations across standards might help readers navigate the data more easily.

Miranda et al., present a timely and relevant investigation into the emergence of Candida auris in Portugal, combining clinical data with genomic and antifungal susceptibility analyses. Despite its relevance, the manuscript must be improved for publication in JoF

Major concerns

1- The identification of a novel missense mutation (I1465L) in the FKS1 hotspot region is a potentially significant finding, given its location in a conserved domain of the β-1,3-glucan synthase. However, the manuscript does not provide functional evidence to support its role in echinocandin resistance.  Even if experimental validation is beyond the scope of this study, a more thorough discussion of the mutation’s potential impact and its novelty in the context of existing literature would strengthen the manuscript.

2- The manuscript presents MIC data interpreted using multiple standards (CLSI, CDC tentative breakpoints, EUCAST ECOFFs), which may confuse readers. Some isolates are described as “non-wild-type” under EUCAST but “susceptible” under CLSI, without a clear rationale for prioritizing one classification over another.

3- The study includes only eight isolates from a single hospital, which limits the generalizability of the findings. While the novelty of the report is appreciated, broader conclusions about transmission, resistance patterns, or clade behavior should be framed cautiously.

Minor concerns

4- Minor grammatical and stylistic issues are present throughout the manuscript.  For example, Introduction: Consider rephrasing “pressing need for global surveillance” to “urgent need for enhanced global surveillance.”

    A thorough proofreading pass is recommended to improve flow and readability.

5- Table 3 (MICs and Susceptibility Profiles): The table is dense and includes multiple interpretive standards (CLSI, ECOFF, TBP). A simplified version or a supplementary table comparing interpretations across standards might help readers navigate the data more easily.

Author Response

Point-by-point response to Reviewer 2

 Miranda et al., present a timely and relevant investigation into the emergence of Candida auris in Portugal, combining clinical data with genomic and antifungal susceptibility analyses. Despite its relevance, the manuscript must be improved for publication in JoF.

Major concerns

Comment 1: The identification of a novel missense mutation (I1465L) in the FKS1 hotspot region is a potentially significant finding, given its location in a conserved domain of the β-1,3-glucan synthase. However, the manuscript does not provide functional evidence to support its role in echinocandin resistance.  Even if experimental validation is beyond the scope of this study, a more thorough discussion of the mutation’s potential impact and its novelty in the context of existing literature would strengthen the manuscript.

Response 1: Thank you very much for your comment. Accordingly to another reviewer's suggestion, we carried out the blast of the genome of our isolates with the genome of the Clade I reference strain B8441. When preparing the manuscript, the reference strain B8441 was not yet available in GenBank and therefore could not be included in our analysis. The genome of this strain was annotated on 22 April 2024 and only released in August 2024, by which point the manuscript had already been completed. We acknowledge, however, that we should have verified whether additional strains (including the reference strain) had become available during the review process. In line with the reviewer’s suggestion, we have included strain B8441 in the analysis and provided a table detailing the variants identified in our strains (Table 5).

Comment 2: The manuscript presents MIC data interpreted using multiple standards (CLSI, CDC tentative breakpoints, EUCAST ECOFFs), which may confuse readers. Some isolates are described as “non-wild-type” under EUCAST but “susceptible” under CLSI, without a clear rationale for prioritizing one classification over another.

Response 2: We agreed that the information regarding MIC was confusing. MICs were determined for each isolate following European Committee on Antimicrobial Susceptibility Testing E.Def.7.4 (EUCAST) (Table 4) and according to M27-A3 protocol of the Clinical Laboratory Standard Institute (CLSI) (Table S3 -Supplementary material) protocols and breakpoints. Findings were separated, and we opted to describe MICs and susceptibility profiles of our isolates according to EUCAST guidelines in the main text (Table 4). At the same time, results from the CLSI protocol can be consulted in the supplementary material (Table S3).

Comment 3: The study includes only eight isolates from a single hospital, which limits the generalizability of the findings. While the novelty of the report is appreciated, broader conclusions about transmission, resistance patterns, or clade behaviour should be framed cautiously.

Response 3: Thank you very much for your comment. Main changes in the manuscript were carried out according to your suggestions (lines 492-507).

Minor concerns

Comment 4: Minor grammatical and stylistic issues are present throughout the manuscript.  For example, Introduction: Consider rephrasing “pressing need for global surveillance” to “urgent need for enhanced global surveillance.” A thorough proofreading pass is recommended to improve flow and readability.

Response 4: Changes were made as suggested (line 61). As a result of the genomes' comparison with the genome of the Clade I reference strain B8441, the manuscript underwent major changes and significant improvements. Furthermore, the manuscript was revised using an English writing assistance software to minimise our written English.

Comment 5: Table 3 (MICs and Susceptibility Profiles): The table is dense and includes multiple interpretive standards (CLSI, ECOFF, TBP). A simplified version or a supplementary table comparing interpretations across standards might help readers navigate the data more easily.

Response 5: Table 3 was reformulated, simplified, and replaced by Table 4. Findings were separated, and we opted to describe MICs and susceptibility profiles of our isolates according to EUCAST guidelines in the main text (Table 4). At the same time, results from the CLSI protocol can be consulted in the supplementary material (Table S3).

Reviewer 3 Report

This manuscript provides valuable insights into the genomic characterization of Candida auris in Portugal. The identification of novel mutations, particularly in the FKS1 gene, is an important contribution to the growing body of literature on C. auris resistance mechanisms. However, the manuscript requires significant revisions to improve clarity.

The title is misleading. Similar studies have already been published, including:

  • Henriques J, Mixão V, Cabrita J, Duarte TI, Sequeira T, Cardoso S, Germano N, Dias L, Bento L, Duarte S, Veríssimo C. Candida auris in intensive care setting: the first case reported in Portugal. Journal of Fungi. 2023, 9(8):837.
  • Nascimento T, Inácio J, Guerreiro D, Patrício P, Proença L, Toscano C, Diaz P, Barroso H. Insights into candida colonization in intensive care unit patients: A prospective multicenter study. Journal of Fungi. 2024, 10(6):378.

Line 30, as mentioned above, there is no need to mention “first report”, as similar studies have already been published in Portugal.

Line 35, Provide the full names for CLSI and EUCAST.

Line 36-38, The phrase "one strain" is unclear. Which strain is being referred to? The author should provide the detailed strain ID.

Line 56-58, The author should explain why these strains are difficult to analyze phylogenetically.

Line 69-70, Are there no reports on Clades V and VI? If not, this should be clarified.

The section on antifungal resistance mechanisms is well-articulated, explaining how C. auris adapts to survive under antifungal pressure. However, it could be enhanced by providing specific examples of the adaptive mechanisms in action. Are there any studies or known cases where these mechanisms have directly contributed to outbreaks or resistance issues? Including examples would strengthen the argument.

The introduction currently presents a broad overview of C. auris, but a brief mention of why this study is important and how it contributes to the field would make it more impactful.

Line 115, It might be helpful to explain the choice of the reference genome from Japan and how it aligns with other global strains of C. auris. Providing reasoning behind the selection of this reference genome will add credibility to the analysis.

Line 292-300, This section of the discussion needs clarification. How is it related to whole genome sequencing and analysis? The connection between the two is not immediately clear.

Author Response

Point-by-point response to Reviewer 3

This manuscript provides valuable insights into the genomic characterization of Candida auris in Portugal. The identification of novel mutations, particularly in the FKS1 gene, is an important contribution to the growing body of literature on C. auris resistance mechanisms. However, the manuscript requires significant revisions to improve clarity.

Comment 1: The title is misleading. Similar studies have already been published, including:

Response 1: Considering the novel findings that resulted from genome blast to the Clade I reference strain (B8441), as suggested by another reviewer, the authors changed the title to “First report of Candida auris candidemia in Portugal: Genomic characterisation and antifungal resistance-associated genes analysis”. In fact, we consider that it is the first report of candidemia in Portugal, once:

Henriques J, Mixão V, Cabrita J, Duarte TI, Sequeira T, Cardoso S, Germano N, Dias L, Bento L, Duarte S, Veríssimo C. Candida auris in intensive care setting: the first case reported in Portugal. Journal of Fungi. 2023, 9(8):837.

This case report describes a patient from Angola who had a positive SARS-CoV-2 antigen test with mild symptoms and was transferred to a Lisbon hospital for liver transplantation, as mentioned in the manuscript. C. auris was isolated from a bronchoalveolar lavage, 4 days before the detection of Acinetobacter baumanii in the same type of biological sample, and under treatment with colistin.

Nascimento T, Inácio J, Guerreiro D, Patrício P, Proença L, Toscano C, Diaz P, Barroso H. Insights into Candida colonisation in intensive care unit patients: A prospective multicenter study. Journal of Fungi. 2024, 10(6):378.

Describes a prospective study aimed to characterise the prevalence of Candida species in the skin of patients admitted at several time points in the ICU of several hospitals in the Lisbon region. Of the 329 positive samples for fungi isolation, none were identified as Candida auris.

Comment 2: Line 30, as mentioned above, there is no need to mention “first report”, as similar studies have already been published in Portugal.

Response 2: For the reasons mentioned in our previous answer, we believe that our study is the first report describing C. auris candidemia, genomic and antifungal susceptibility profile characterisation.

Comment 3: Line 35, Provide the full names for CLSI and EUCAST.

            Response 3: Full names were added as suggested (lines 38 and 160-161).

Comment 4: Line 36-38, The phrase "one strain" is unclear. Which strain is being referred to? The author should provide the detailed strain ID.

Response 4: The phrase has been deleted.

Comment 5: Line 56-58, The author should explain why these strains are difficult to analyse phylogenetically.

Response 5: Thank you very much for your comment. We have clarified this point in the revised manuscript (lines 65-70).

Comment 6: Line 69-70, Are there no reports on Clades V and VI? If not, this should be clarified.

Response 6: Regarding Clade V, it was first reported on single isolates from Iran (please see reference 9). Regarding Clade VI, two independent reports (Bangladesh and Singapore) have been characterised in 2024, isolates that cluster apart from known clades, supporting a novel sixth clade. (Please see references 8 and 10.) 

Comment 7: The section on antifungal resistance mechanisms is well-articulated, explaining how C. auris adapts to survive under antifungal pressure. However, it could be enhanced by providing specific examples of the adaptive mechanisms in action. Are there any studies or known cases where these mechanisms have directly contributed to outbreaks or resistance issues? Including examples would strengthen the argument.

Response 7: Yes, there are studies where specific resistance mutations (especially FKS1 for echinocandins and ERG11/TAC1B for azoles) have directly contributed to outbreak propagation, treatment failures, and the emergence of pan-resistant clusters in healthcare settings. Several examples were described in the discussion section (lines 430-492) (references 33, 36, 061, 62).

Comment 8: The introduction currently presents a broad overview of C. auris, but a brief mention of why this study is important and how it contributes to the field would make it more impactful.

Response 8: A sentence explaining this study's aims (lines 90- 95) was introduced.

Comment 9: Line 115, It might be helpful to explain the choice of the reference genome from Japan and how it aligns with other global strains of C. auris. Providing reasoning behind the selection of this reference genome will add credibility to the analysis.

Response 9: Thank you for this observation. Initially, we selected the Japanese reference genome because at the time of our analysis, it was the most complete and readily available assembly in GenBank, and it has been widely used in previous studies of C. auris. However, following the reviewer’s comments, given that strain B8441 (Clade I, corresponding to our isolates) has since been released and annotated, we have updated our analysis accordingly. Using B8441 as the reference provides a more appropriate genomic context for our isolates, ensuring higher accuracy and relevance of the variant calling results. We have clarified this point in the revised Methods section and Discussion.

Comment 10: Line 292-300, This section of the discussion needs clarification. How is it related to whole-genome sequencing and analysis? The connection between the two is not immediately clear.

Response 10: The introductory paragraph of the Discussion has been revised to clarify the connection between colonisation, skin microbiota, and whole-genome sequencing (WGS). We now explicitly state that WGS enables detailed characterisation of C. auris strains, providing insights into their genetic diversity, antifungal resistance profiles, and potential transmission pathways, thus linking epidemiological observations with genomic analysis.

Round 2

Reviewer 1 Report

Comment 1: The title is somehow misleading, since there was a report of the first colonization in Portugal in 2022. Consider revising the title; e.g. “Genomic characterization of C. auris candidemia in Portugal reveals novel FKS1 mutation and…”. Alternatively, remove the word “colonization”, if this is the first case of candidemia in Portugal.

Response 1: Thank you very much for your suggestion. Considering the novel findings that resulted from genome blast to the Clade I reference strain (B8441), as suggested by the Reviewer, the authors changed the title to “First report of Candida auris candidemia in Portugal: Genomic characterisation and antifungal resistance-associated genes analysis”. The word colonization has been removed.

Thank you for the clarification. 

Comment 2: There is some issues and/or information lacking from the result section “Genetic variants detection”. Which reference genome was used for alignment and analysis? In Figure 4, strain B11220 is used as a reference. However, this is a clade II strain. The reference strain (e.g., on Candida Genome Database) for C. auris genomic analysis is strain B8441 of Clade I (same clade as isolates of this study), which should also be used for analysis. Please also discuss synonymous vs nonsynonymous variants. Also, compare variants between strains of this study, and with a closely related strain as reference (like B8441, or closer), to identify variants of interest in genes other than those investigated in the context of drug resistance.

Response 2: Thank you for your comment. When preparing the manuscript, the reference strain B8441 was not yet available in GenBank and therefore could not be included in our analysis. The genome of this strain was annotated on 22 April 2024 and only released in August 2024, by which point the manuscript had already been completed. We acknowledge, however, that we should have verified whether additional strains (including the reference strain) had become available during the review process. In line with the reviewer’s suggestion, we have included strain B8441 in the analysis and provided a table detailing the variants identified in our strains (Table 5, Figures 2 and 3, Supplementary Material Table S1).

The B8441 reference genome is available since 2017; https://www.ncbi.nlm.nih.gov/datasets/genome/GCA_002759435.2/

The latest version is from 2024, and the isolates of this manuscript are from 2023, so I don’t really understand this argument. Since this annotated reference has been around for 8 years and is used in close to all Candida auris genomic analyses to date, it is the most recommended reference sequence to use, so thank you for including this reference in the analysis.

Comment 3: The reported FKS1 mutation does not match the reference sequence. In the officially annotated sequence of FKS1 of C. auris, available here: http://www.candidagenome.org/cgi-bin/locus.pl?locus=B9J08_000964, the nucleotide at position 4393 is a Thymine (T), and the codon translates a Tryptophan (W), unlike the reported Adenine (A) translating an Isoleucine (I). This major issue might be the result of the use of a wrong reference sequence (see second comment), and the reported mutation might solely be a inter-clade substitution! In addition to being wrongly annotated, the FKS1 hotspot 1, is very far away from the reported mutation. Hotspot 1 spans nucleotides 1903-1929, nowhere near the reported “hotspot 1”. In fact, the sentence “In the gene encoding the β-1,3-glucan synthase FKS1 hotspot1 (HS1), a novel nucleotide 242 substitution at position A4393C resulted in the amino acid substitution I1465L (Figure 4).” is wrong, as “hotspot 1” is not a gene, it’s a region in the gene. I suggest revision of the whole genome analysis data by a genomic analysis expert.

Response 3: Thank you for your comment. We understand the reviewer’s concern regarding using an incorrect reference sequence. As mentioned in our previous response, the reference strain B8441 was unavailable during manuscript preparation. However, as suggested, we have included the Clade I reference strain (B8441) in our analysis. Upon performing the FKS1 gene alignments, we found that the previously reported FKS1 mutation does not match the reference sequence. Consequently, we removed Figure 4 and replaced it with a table summarising the nucleotide and protein variants identified in our strains (Table 5).  

Thank you for rectifying this. It is intriguing that the strain is echinocandin resistant, without any FKS mutation.

In line 443, you write; “In our isolates, no 443 nonsynonymous variants were detected in CIS2. “, but you do identify a CIS2 missense mutation? Also, note that reference 35 (supposed to be https://journals.asm.org/doi/full/10.1128/mbio.03333-20) is wrongly cited. Please revise the references to make sure no other issues are present.  

Comment 4: Often, there is info missing regarding analyses and software in the methods section. For example, the variant calling algorithm is not described in detail, nor referenced. Which algoritm and program were used? Or in the case of “All 160 mutations were confirmed by visually inspecting the alignment of the reads to the corresponding assemblies.” How were they visually inspected? Using IGV on the alignment? Please provide more detailed methods.

Response 4: Thank you for your comment. We have clarified and expanded the Methods section to provide detailed information on detecting and confirming antifungal resistance gene mutations. Specifically, we now describe the variant calling algorithm used, the annotation and translation of variants into amino acid substitutions, and the procedure for visual confirmation of all mutations using Integrative Genomics Viewer (lines 170-183).

OK

Comment 5: The use of crude amount of SNPs as indicator of relatedness (line 318) is not okay. Please use reference methods such as multi-locus sequence typing (e.g. https://journals.asm.org/doi/full/10.1128/mbio.02971-19) or phylogenomics to type the strains.

Response 5: We thank the reviewer for this valuable comment. We agree that crude SNP counts alone cannot infer genetic relatedness among strains. In the revised manuscript, we have clarified that the reported SNP differences cannot be used as definitive evidence of relatedness. We also highlight that future analyses using multi-locus sequence typing (MLST) or core genome phylogenomics would be necessary to accurately assess these isolates' genetic relationships and potential transmission pathways (lines 373-380).

 OK

Comment 6: Line 71-76; following study might be a better/extra reference for that statement: https://doi.org/10.1038/s41564-024-01854-z

Response 6: The reference was added (line 88).

 OK

Comment 7: Table 1 is too big and out of proportion.

Response 7: In the submitted manuscript version, Table 1 is now Table 2 and has been redimensioned.

 OK

Comment 8: Line 193: “although death was not exclusively attributed to C. auris infection.” Could you report cause of death?

Response 8: In fact, it would be interesting to point to sepsis as the cause of death; however, these patients, besides candidemia, displayed a complex state of disease, including neoplasia. Therefore, it would be incorrect to attribute candidemia as the cause of death.

 OK

Comment 9: There is confusion regarding CLSI vs EUCAST. Which method was used? Why are both mentioned?

Response 9: MICs were determined for each isolate following EUCAST (Table 4) and CLSI (Table S3 - Supplementary material) protocols and breakpoints. Findings were separated, and we opted to describe MICs and susceptibility profiles of our isolates according to EUCAST guidelines in the main text (Table 4). At the same time, results from the CLSI protocol can be consulted in the supplementary material (Table S3).

 OK

Comment 10: How were the genes chosen that were investigated in “Analysis of resistance-associated mutations”? 3

Response 10: The genes are summarised in Table 1. The drug resistance-associated genes were chosen according to previous reports and are involved in known fungal resistance mechanisms to antifungal drugs: the ergosterol biosynthesis pathway, efflux pumps and their transcription factors, echinocandins targets (FKS genes), epigenetic regulators, and others.

 OK

Comment 11: Table 3 is confusing and an unusual way of presenting this sort of data. Consider choosing another way of presenting it. In addition, please provide the susceptibility profiles (OD Measurements in function of drug concentration), a supplementary material. I am especially interested in the profile for caspofungin, as the eagle effect is known to distort MIC interpretations. With a strain being resistant to caspofungin and not micafungin or anidulafungin, supra MIC growth (tolerance) or the eagle effect might play a role in strain 275.

Response 11: Table 3 was reformulated, simplified, and replaced by Table 4. MIC values were determined according to the European Committee on Antimicrobial Susceptibility Testing E.Def.7.4 (EUCAST) protocol. Despite high MIC levels to caspofungin, no eagle effect was observed.

 OK

Comment 12: What is the relevance of the start of the discussion (lines 292-300)? Please remove information that is irrelevant to the manuscript.

Response 12: Thank you very much for your comment. The discussion was changed, and inadequate information was removed.

 OK

None

Reviewer 2 Report

The authors have successfully addressed my questions/concerns

The authors have successfully addressed my questions/concerns

Reviewer 3 Report

The author revised the manuscript according to the reviewer's comments.

The author revised the manuscript according to the reviewer's comments.